# Discovery of a Unique Set of Dog-Seroreactive *Coccidioides* Proteins Using Nucleic Acid Programmable Protein Array

**DOI:** 10.3390/jof10050307

**Published:** 2024-04-24

**Authors:** Megan A. Koehler, Lusheng Song, Francisca J. Grill, Lisa F. Shubitz, Daniel A. Powell, John N. Galgiani, Marc J. Orbach, Edward J. Robb, Yunro Chung, Stacy A. Williams, Vel Murugan, Jin-gyoon Park, Joshua LaBaer, Douglas F. Lake, D. Mitchell Magee

**Affiliations:** 1School of Life Sciences, Arizona State University, Tempe, AZ 85287, USA; makoehle@asu.edu; 2Biodesign Institute, Arizona State University, Tempe, AZ 85287, USA; himsong@asu.edu (L.S.); yunro.chung@asu.edu (Y.C.); stacy.adriana.williams@asu.edu (S.A.W.); velmurugan@asu.edu (V.M.); jin.park.1@asu.edu (J.-g.P.); jlabaer@asu.edu (J.L.); 3Cactus Bio, LLC, Phoenix, AZ 85259, USA; fgrill@asu.edu; 4Valley Fever Center for Excellence, The University of Arizona, Tucson, AZ 85724, USA; lfshubit@arizona.edu (L.F.S.); danielpowell@arizona.edu (D.A.P.); spherule@arizona.edu (J.N.G.); orbachmj@arizona.edu (M.J.O.); 5BIO5 Institute, The University of Arizona, Tucson, AZ 85724, USA; 6Department of Immunobiology, The University of Arizona, Tucson, AZ 85724, USA; 7Department of Medicine, The University of Arizona, Tucson, AZ 85724, USA; 8School of Plant Sciences, The University of Arizona, Tucson, AZ 85724, USA; 9Anivive Life Sciences, LLC, Long Beach, CA 90807, USA; edward@anivive.com; 10College of Health Solutions, Arizona State University, Phoenix, AZ 85004, USA

**Keywords:** coccidioidomycosis, valley fever, dogs, antibodies, diagnostics

## Abstract

Valley Fever (VF), caused by fungi in the genus *Coccidioides*, is a prevalent disease in southwestern and western parts of the United States that affects both humans and animals, such as dogs. Although the immune responses to infection with *Coccidioides* spp. are not fully characterized, antibody-detection assays are used in conjunction with clinical presentation and radiologic findings to aid in the diagnosis of VF. These assays often use Complement Fixation (CF) and Tube Precipitin (TP) antigens as the main targets of IgG and IgM reactivity, respectively. Our group previously reported evidence of over 800 genes expressed at the protein level in *C. posadasii*. However, antibody reactivity to the majority of these proteins has never been explored. Using a new, high-throughput screening technology, the Nucleic Acid Programmable Protein Array (NAPPA), we screened serum specimens from dogs against 708 of these previously identified proteins for IgG reactivity. Serum from three separate groups of dogs was analyzed and revealed a small panel of proteins to be further characterized for immuno-reactivity. In addition to CF/CTS1 antigen, sera from most infected dogs showed antibody reactivity to endo-1,3-betaglucanase, peroxisomal matrix protein, and another novel reactive protein, CPSG_05795. These antigens may provide additional targets to aid in antibody-based diagnostics.

## 1. Introduction

Valley Fever (VF), also called coccidioidomycosis, is the disease caused by the dimorphic fungal species *Coccidioides immitis* and *Coccidioides posadasii* [1]. *Coccidioides* spp. grow in arid soils, mainly in the southwestern deserts of North America as well as desert regions in Central and South America [2]. With the increase in global temperatures, the possibility of *Coccidioides* becoming endemic in more regions has increased [3]. Unlike an opportunistic pathogen, *Coccidioides* spp. are primary pathogens and can cause disease in otherwise healthy individuals, most commonly observed in humans and dogs [4]. In one cross-sectional study evaluating *Coccidioides* infection risk factors among dogs living in an endemic area (Tucson, AZ), 32 of 381 dogs had anti-coccidioidal antibodies, of which 13 were clinically ill and 19 were asymptomatic despite positive serology [4]. While active infection can only be definitively diagnosed by the isolation or identification of *Coccidioides* spp. from a clinical specimen, antibody-based assays remain important as diagnostic aids.

Antigen preparations used for serodiagnosis are often complex fungal extracts. Two key systems that are used in the diagnosis of disease are the Tube Precipitin (TP) or the Complement Fixation (CF) antigens [5]. As a sign of early infection, IgM is detected by reactivity to a complex, high-molecular-weight (120–330 kDa), heat-stable carbohydrate antigen [6,7]. A glycoprotein component of TP was identified as beta-glucosidase (Bgl2) [8]. Originally defined as an antigen that fixes complement, the Complement Fixation (CF) antigen is used to detect IgG reactivity [5]. More recently, this antigen has been shown to be proteinaceous and has been molecularly cloned and identified with enzymatic endochitinase reactivity (CTS1), further noted in this study as CF/CTS1 [9,10]. As noted above, antibody-detection assays primarily utilize fungal extracts as antigenic preparations. These extracts are heterogenous in content and are difficult to standardize from batch to batch. Additionally, laboratories and commercially available kits use antigen preparations that are prepared differently with no stated quantitation of an immunoreactive component, which may cause a variation in diagnosis and monitoring [11,12]. We propose that a defined, recombinant, multi-antigen test could increase the accuracy of testing and possibly allow the detection of disease earlier than current testing allows. In order to identify seroreactive antigens for a multi-antigen test, high-throughput screening for antibody reactivity against the annotated *Coccidioides* proteome is required [13].

A useful assay for screening large numbers of proteins against serum in a high-throughput manner is the Nucleic Acid Programmable Protein Array (NAPPA) [14]. This technology allows for large sets of proteins, such as the *Coccidioides* proteome, to be screened against large sets of serum specimens. NAPPA works by using protein microarrays on a slide. DNA plasmids are printed into individual wells on the slide followed by the addition of in vitro transcription and translation mixtures to allow for the proteins to be made and captured for display in the individual wells. Serum specimens are then allowed to incubate with the array, and antibodies present in the sera for individual proteins can be detected using fluorescent secondary antibodies. In this manner, NAPPA screening can identify potentially seroreactive proteins from a large antigen set. These potential seroreactive proteins will then be characterized in depth in orthogonal assay systems, such as ELISA.

Using the *Coccidioides* proteome previously reported by our group [13], we used NAPPA to reveal the landscape of serologic reactivity to 708 coccidioidal proteins from dogs in the following groups: (i) naturally infected in the endemic area, (ii) infected with a laboratory strain of *C. posadasii*, and (iii) vaccinated with a Δ*cps1* live attenuated *C. posadasii* strain followed by challenge with the same laboratory strain, allowing for the discovery of unique seroreactive proteins not previously identified as immunogenic targets.

## 2. Materials and Methods

### 2.1. NAPPA

For this project, NAPPAs (708 *Coccidioides* genes on each array) were used to screen sera. The DNA was ordered through Genscript (Piscataway NJ, USA) in pANT7_cGST, with the selected DNA being from proteomic work previously performed by our group [13]. To summarize the methods below, expression plasmid DNA containing the individual open reading frames as a fusion construct with glutathione-S transferase as a tag (pANT7-GST parent vector) was admixed with anti-GST antibody in an albumin cross-linked matrix and printed onto nano-well silicon slides. The clones are available from DNASU (https://dnasu.org/DNASU/Home.do, accessed on 10 March 2024). On the day of assay, the *Coccidioides* proteins were produced by addition of an HeLa-cell-derived transcription/translation mix (Thermo Fischer Scientific, Waltham, MA, USA) for protein expression as fusion constructs with the GST tag. After protein expression, the slide was immediately probed with serum specimens followed by washing and the addition of secondary goat anti-canine IgG coupled to a fluorochrome. After washing excess secondary antibody from the slides, they were imaged and quantified to identify coccidioidal proteins that were seroreactive to serum specimens from infected or vaccinated and infected dogs. These methods are outlined in further detail by Song et al. [14]. Quality control measures are assessed on each print to include the minimum level of protein expression so that 96% of the targeted proteins are produced.

### 2.2. Serum Specimens

Serum specimens screened by NAPPA included single time-point specimens from (i) client-owned dogs residing in VF-endemic areas (non-endemic region samples as negative controls), (ii) longitudinal specimens from laboratory-infected dogs (range—3300 to 1 × 10^6^ arthroconidia intratracheally dogs [15], and (iii) dogs vaccinated with a ∆cps1 mutant strain of *C. posadasii* followed by intratracheal challenge with 10,000 arthroconidia of a virulent laboratory strain [15]. The laboratory-obtained specimens were donated to our group for this study. Specimens from dogs residing in endemic and non-endemic areas were obtained from surplus dog sera from other diagnostic investigations in our laboratory and were not specifically collected for this study. This included coccidioidomycosis-negative sera from 17 dogs residing in a VF-non-endemic region (VA, USA) and sera from 17 dogs residing in Maricopa County, USA, a VF-endemic region. For the endemic sera, 10 specimens came from dogs with serologic evidence of a natural infection while the other 7 had no serologic evidence of infection and were used as negatives. Serologic evidence was defined as IgG reactivity by immunodiffusion.

Longitudinal specimens from laboratory-infected dogs were donated from Shubitz et al., at the University of Arizona. These serum specimens were obtained from purpose-bred beagles (N = 18) infected with the virulent laboratory strain *C. posadasii* Silveira [15]. From this group, we analyzed 106 longitudinal serum specimens from blood drawn at 2, 4, 6, and 8 weeks post infection.

The final subset of dog sera donated included 54 specimens longitudinally collected from 6 dogs that were vaccinated twice 28 days apart with 100,000 arthroconidia of the Δ*cps1* vaccine, and then challenged with 10,000 arthroconidia of WT strain Silveira in week 10 [15]. Serum was collected prior to vaccination (V0) and at 2, 4, 6, and 10 weeks post vaccination and then every 2 weeks post infection until euthanasia at 56 days (9 samples per dog) [15].

### 2.3. Data Analysis

Fluorescence intensity raw data values from each spot on the slides were quantified and normalized to the median fluorescence intensity of each subarray, which was then called the median normalized intensity (MNI). Antibody-reactive proteins were defined as a fluorescence value greater than or equal to 1.975, determined by using an ROC curve for the highest possible sensitivity (80%) and specificity (97.87%), based on CF/CTS1 values for known negative and positive dogs, as CF/CTS1 is the main antigen in most diagnostic tests [5]. Prism (version 10.2.2) and Excel (version 2403, build 17425.20176) were used to sort, analyze, and graph data. Percent positivity for each *C. posadasii* protein was calculated by dividing the number of positive seroreactive antigens by the total number of serum specimens in each group. Odds ratios were calculated by using the serological responses from negative specimens, such as pre-bleeds and known negative cohorts, as compared to each representative positive timepoint, such as the post-challenge or post-vaccine time point, to measure the association between the outcomes of positive and negative data sets [16]. In addition to the odds ratio, the *p*-value was calculated by performing a Chi-square test on each group’s data set, and a *p*-value cutoff of 0.05 was used for the determination of significance [16], as shown in Appendix A.

## 3. Results

### 3.1. Overlapping Proteins between the Different Groups

#### 3.1.1. Laboratory and Naturally Infected Dogs

A total of 708 Coccidioidal proteins were screened against the three different groups of dog sera, including laboratory-infected dogs (N = 18); dogs vaccinated with live, avirulent *C. posadasii* arthroconidia and then infected in the laboratory (N = 6); and naturally exposed dogs with serologic evidence of infection from an endemic area (Maricopa County, AZ, USA) (N = 10). Using these three different groups, sero-reactivity to the proteins was ranked using the median normalized intensity (MNI) to each protein. As a screening selection strategy, we calculated the odds ratio and determined 2.0 to be the cutoff for selecting differential seroreactive proteins between negative and positive groups. Using this odds ratio, the laboratory-infected dogs revealed a total of one-hundred and eighty-six seroreactive proteins. For the naturally infected dogs, a total of sixty-four seroreactive proteins were revealed. In Figure 1, we show thirty-three proteins as an overlap between laboratory-infected dogs and naturally infected dogs.

#### 3.1.2. Laboratory-Infected, Naturally Infected, and Vaccinated and Challenged Dogs

The third group were dogs vaccinated with Δ*cps1* followed by challenge with *C. posadasii* Silveira (N = 6). In this group, a total of sixteen proteins were revealed. Vaccination prevented severe disease, potentially minimizing an antibody response against many proteins that were observed when dogs were infected in the laboratory or naturally. When compared against both laboratory-infected and naturally infected dogs, a total of eight unique proteins were identified that were reactive at any timepoint among all groups with an odds ratio >2 shown in Figure 2.

### 3.2. Longitudinal Analysis

#### 3.2.1. Laboratory-Infected Dogs

We also evaluated the longitudinal data in the laboratory-infected dogs (N = 18) since they were given quantified doses of *C. posadasii* Silveira. Selecting seroreactive proteins that were positive longitudinally at a minimum of three timepoints led to seven unique highly seroreactive proteins, as shown in Table 1. The proteins are listed in the table in the order they first became seroreactive, with all proteins being negative in pre-bleed serum specimens as well as the other negative cohorts.

#### 3.2.2. Vaccinated and Challenged Dogs

Further analysis of longitudinal specimens from the “vaccinateds and challenged” (N = 6) canine group revealed three proteins that were highly seroreactive and overlapped with the other laboratory-infected dogs [Table 2]. Interestingly, only hypothetical protein CPSG_05795 was seroreactive starting at two weeks post vaccination and remained highly seroreactive throughout this study.

## 4. Discussion

Using sera from three different cohorts of *Coccidioides*-infected dogs, we employed NAPPA to screen the spherule-based proteome of *Coccidioides* [13] and to determine which coccidioidal proteins were seroreactive. After screening 708 proteins, 33 proteins were identified that are reactive in both laboratory-infected dogs, and naturally infected community dogs with serologic evidence of coccidioidomycosis. Although we cannot determine the species or strains of *Coccidioides* to which naturally infected dogs from the endemic area were exposed, the proteins we identified were reactive with serum from both sets of dogs, suggesting that these 33 proteins are immunogenic across different strains of the fungus. Since the NAPPA process here primarily aims to screen a large set of proteins for further validation with other cohorts, we focused on the seropositive antigens in the three coccidioidomycosis-positive cohorts (>2 odds ratio) and not reactive in the coccidioidomycosis-negative cohort. In the vaccinated dogs, there was not as wide of a range of reactivities, potentially caused by vaccination preventing severe disease, thus reducing antibody reactivity. However, in the 33 reactive proteins between lab- and naturally infected, eight proteins were also observed as being highly reactive in vaccinated dogs before challenge. These dogs were vaccinated with a gene-deletion mutant, Δ*cps1*, of strain Silveira, and the eight proteins stimulated an antibody response prior to challenge with the virulent parental strain. It appears that these eight proteins may be strong and early markers of exposure to *Coccidioides*, whether as a vaccine or in a pulmonary challenge. For the naturally infected dogs from the community, a drawback is that the time of their infection is unknown and there were no serial samples to analyze, but the eight proteins were also seroreactive in these animals, which could further support the durability of these proteins as diagnostic disease markers.

To identify coccidioidal antigens that might be included in a multi-antigen diagnostic test, we evaluated proteins that were highly seroreactive longitudinally, determining how early the response appeared in laboratory-infected dogs and whether the seroreactivity was durable over time. Of the seven highly reactive proteins identified at three of four timepoints between 2 and 8 weeks post infection in the laboratory dogs, three were highly reactive in week 2, before CF reactivity appeared, which is notable since CF/CTS1 antigen is routinely used for the commercial testing of IgG antibodies in dogs for the diagnosis of VF. Out of these three early proteins, the “hypothetical protein CIMG_05795” continued to demonstrate strong seroreactivity throughout each time point, where reactivity to the other proteins appeared transient. This protein requires additional investigation and further testing in more dogs to verify that this protein may identify infection earlier than the CF/CTS1 antigen in dogs.

The proteins identified as longitudinally seroreactive in this study show promise as potential new diagnostic markers. While most VF diagnostics rely on reactivity to one antigen alone, typically CF/CTS1, the additional seroreactive proteins identified by NAPPA can potentially be used in conjunction to increase sensitivity and specificity. Using a multi-antigen diagnostic, especially with a protein such as “uncharacterized protein CPSG_05795”, detection may be possible in week 2, as opposed to weeks 4 and 6 as observed with CF/CTS1. Benefits of detecting infection earlier, perhaps when symptoms are acute, include a possible earlier therapeutic intervention and improved antimicrobial stewardship.

In this study, we discovered a new seroreactive protein, “uncharacterized protein CPSG_05795”, which was previously discovered by our group in a proteomic study [17]. However, other seroreactive proteins identified in this study have been previously described. Endo-1,3-betaglucanase is a protein involved in the pathogenesis of fungi in plants and may potentially be involved in human pathogenicity. It is a component of the fungal wall and is important in degradation of the cell wall [18,19]. It is known to bind and activate C-type lectin receptors, specifically Dectin-2 receptors, initiating the CARD9 signaling pathway [20,21]. This pathway then initiates Th1 and Th17 immune pathways, which are crucial pathways for anti-fungal immunity, including *Coccidioides* infection. In addition, Dectin-2 deletion results in a higher susceptibility of mice to fungal disease [20,21]. Not only was the Endo-1,3-beta-Glucanase shown to be a highly seroreactive protein induced by vaccination, further work building off the current literature [20,22] will be performed to evaluate Endo-1,3-beta-Glucanase as a T-cell marker as well, which could be used as a marker of a strong immune response to the vaccine. Peroxisomal matrix protein (Pmp1) was another highly reactive antigen identified in this study. Pmp1 has been identified in previous proteomic studies and was also used as a potential vaccine candidate, showing protection in murine models [23]. The identification of Pmp1 by both studies shows the importance of sero-screening to identify highly reactive proteins for potential use in diagnostics and beyond.

While CPSG_05795 is uncharacterized, there is proteomic [13] and transcriptomic evidence for its expression. It is a proline-rich protein, but unrelated to the proline-rich antigen/Ag2 vaccine candidate [24,25,26] which appears to be unique to *Coccidioides*. CPSG_05795 does have a *C. immitis* homolog, CIMG_05576. Further studies of the structure and localization of CPSG_05795 are warranted to identify its role in *Coccidioides* biology.

One biological reality and limitation of this study was the relatively small sample population coupled with the heterogeneous immune responses of dogs. Even with a majority of the serum specimens derived from moderately inbred laboratory beagles infected only with *C. posadasii* str. Silveira, the seroreactivity of the dogs varied greatly. We infer from this that additional work needs to be performed with more naturally infected dogs, including animals from California that are more likely to be infected with *C. immitis* strains (probably needs a reference for humans), and if the opportunity arises, with more laboratory-infected dogs to verify and refine the antigens that are most likely to yield a broad-spectrum test that detects disease earlier.

Other limitations were that our analysis focused on IgG reactivity, not IgM reactivity. As such, it is possible that we might have missed very early carbohydrate antigens [27]. However, some of the IgG seroreactive proteins identified by NAPPA appeared as early as two weeks post infection. Additionally, while antibodies are a critical component of the immune system, T cell reactivity and functions were not evaluated. Future studies looking at the T-cell reactivity of these proteins are needed to obtain a better understanding of the immune response induced by these proteins.

## 5. Conclusions

Using NAPPA of 708 coccidioidal proteins, we screened serum from three cohorts of dogs: (1) naturally infected, (2) vaccinated, and (3) vaccinated then challenged. Thirty-three proteins were seroreactive with naturally infected and vaccinated/challenged dogs, and one “hypothetical protein” was seroreactive by week two after vaccination and remained seroreactive up until 56 days post infection, the conclusion of this study. Further analysis and characterization may suggest that this protein be included in a multi-antigen diagnostic test.

## Figures and Tables

**Figure 1 jof-10-00307-f001:**
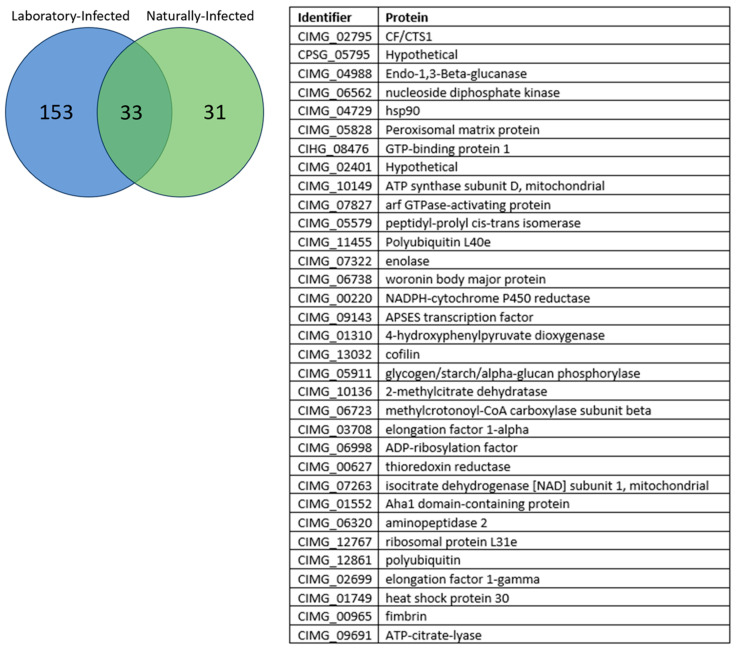
Venn diagram of the thirty-three overlapping seroreactive proteins from the laboratory-infected and seropositive naturally infected dogs. The thirty-three overlapping proteins and their protein identifiers are listed in the table to the right of the diagram.

**Figure 2 jof-10-00307-f002:**
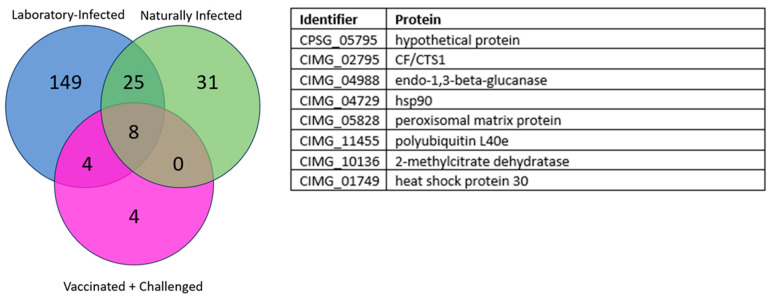
Venn diagram of the eight overlapping seroreactive proteins from all three groups: laboratory-infected (blue circle), sero-positive endemic dogs (green), and vaccine + challenge (pink). The eight overlapping proteins and their protein identifiers are listed in the table to the right of the diagram.

**Table 1 jof-10-00307-t001:** Odds ratio heatmap of laboratory-infected dogs, longitudinally. The table shows the odds ratios for the top seroreactive proteins over time in the laboratory-infected dogs. The heatmap ranges from blue (low odds ratio) to red, with dark red representing odds ratios greater than ten.

		Weeks Post Infection			
Identifier:	Protein:	2	4	6	8
CIMG_06562	nucleoside diphosphate kinase	5	2	2	6
CIMG_01267	26S proteosome regulatory subunit N7	3	2	1	2
CPSG_05795	Hypothetical	9	30	37	18
CIMG_04729	Hsp90	1	4	4	4
CIMG_02795	CF/CTS1	1	41	96	96
CIMG_04988	Endo-1,3-Beta-Glucanase	1	10	24	44
CIMG_05828	Peroxisomal Matrix Protein	1	2	7	16

**Table 2 jof-10-00307-t002:** Odds ratio heatmap of vaccinated and challenged dogs, longitudinally. The table shows the odds ratios for the top longitudinal seroreactive proteins over time for the vaccinated and challenged dogs. The heatmap ranges from blue (low odds ratio) to red, with dark red representing odds ratios greater than ten.

		Weeks Post Vaccine				Weeks Post Infection			
Identifier:	Protein:	2	4	6	10	2	4	6	8
CIMG_02795	CF/CTS1	1	10	20	5	5	10	10	20
CIMG_04988	Endo-1,3-Beta-Glucanase	2	10	20	20	5	10	20	10
CPSG_05795	Hypothetical	10	5	50	20	50	20	50	20

## Data Availability

Data are contained within the article.

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
