# Peer review of "Discovery of a Unique Set of Dog-Seroreactive Coccidioides Proteins Using Nucleic Acid Programmable Protein Array"

_jof, 2024, doi:10.3390/jof10050307_

Round 1

Reviewer 1 Report

Comments and Suggestions for Authors

I appreciate the opportunity to review the manuscript titled 'Discovery of a Unique set of Dog-Seroreactive Coccidioides Proteins Using Nucleic Acid Programmable Protein Array.' Considerable effort has gone into crafting this work. The methods and statistical analysis are meticulously executed, contributing to the overall robustness of the study."

There are just a few aspects the author needs to reconsider:

Line 44 should contain the name of the endemic area.

Omit "Summary" at Line 86.

The manuscript should contain information about the statistical analysis software utilized.

Line 234 contains a typographical error.

Line 264: It seems to lack a necessary reference.

In the range of lines 302 through 311: Due to the involvement of animals in the study, approval from the Institutional Animal Care and Use Committee is necessary.

Author Response

We thank you for taking the time to review our communication, and appreciate your edits and suggestions. We have made the edits as follows:

  1. Line 44 should contain the name of the endemic area.
    1. The endemic area, Tucson, AZ, has been added to the sentence.
  2. Omit "Summary" at Line 86.
    1. Done
  3. The manuscript should contain information about the statistical analysis software utilized.
    1. Added in that Excel and Prism were used for data organization and analysis.
  4. Line 234 contains a typographical error.
    1. Fixed
  5. Line 264: It seems to lack a necessary reference.
    1. Fixed
  6. In the range of lines 302 through 311: Due to the involvement of animals in the study, approval from the Institutional Animal Care and Use Committee is necessary.
    1. We have added in the ethics statement that these samples are exempt from regulatory control, as the work was not done by us as samples were donated to us. We fixed the methods to make this clear.

Reviewer 2 Report

Comments and Suggestions for Authors

Discovery of a Unique set of Dog-Seroreactive Coccidioides Proteins Using Nucleic Acid Programmable Protein Array

 Authors identified new antigens that can provide additional targets to aid in antibody-based diagnostics of Valley Fever.

It is work with a very practical nature, which aims to improve the diagnosis of the disease, which gives it importance in the day-to-day activities of laboratories.

It has some weaknesses, which the authors themselves mention, but overall it achieves the proposed objective.

It is not very clear how the plasmid constructions were made to obtain the 708 proteins; I think that this part of the materials and methods could be improved.

In line 242 of the Discussion the term antibiotic stewardship must be replaced to antimicrobial stewardship, as the work is not about bacteria.

Author Response

We thank you for taking the time to read our communication, and appreciate the comments and suggestions. We have made the suggested edits as follows:

  1. It is not very clear how the plasmid constructions were made to obtain the 708 proteins; I think that this part of the materials and methods could be improved.
    1. we have added into the methods that the plasmids (pANT7_cGST) were ordered from Genscript and designed based off of previous proteomic work
  2. In line 242 of the Discussion the term antibiotic stewardship must be replaced to antimicrobial stewardship, as the work is not about bacteria.
    1. Antibiotic has been changed to antimicrobial